# The Role of Transient Crosslinks in the Chromatin Search Response to DNA Damage

**DOI:** 10.3390/ijms262311697

**Published:** 2025-12-03

**Authors:** Andrew T. Atanasiu, Caitlin Hult, Daniel Kolbin, Benjamin L. Walker, Mark Gregory Forest, Elaine Yeh, Kerry Bloom

**Affiliations:** 1Department of Biology, University of North Carolina at Chapel Hill, Chapel Hill, NC 27599, USAkolbin@live.unc.edu (D.K.); kerry_bloom@unc.edu (K.B.); 2Department of Mathematics, Gettysburg College, Gettysburg, PA 17325, USA; 3Department of Mathematics, University of California Irvine, Irvine, CA 92697, USA; 4Departments of Mathematics, Biomedical Engineering, and Applied Physical Sciences, University of North Carolina at Chapel Hill, Chapel Hill, NC 27599, USA

**Keywords:** DNA damage, SMC complexes, polymer modeling, double-strand breaks, crosslinking

## Abstract

Homology search is a means through which DNA double-strand breaks (DSBs) explore the genome for sequences that enable error-free repair, known as homologous recombination. A better understanding of this search process is fundamental to the relationship between higher-order chromosome organization and DNA damage. Here, we use an entropic bead-spring polymer chain model to simulate the spatiotemporal dynamics of the yeast genome during interphase. The chromosome is organized by transient and dynamic cross-links representing structural maintenance of chromosome (SMC) complexes. DNA damage is modeled as a break in the bead-spring chain, coupled with a removal of crosslinks from beads proximal to the break site. We show that the removal of cross-links drives the exploration of genomic space by the damaged ends, while rates and densities of intact dynamic crosslinking have only a minor role. Local depletion of SMC cross-links proximal to the break site enables the damaged segment to escape the chromosome territory and enhances its ability to explore the genome. Our study reveals a foundational principle by which DSBs can encounter distant regions of sequence homology.

## 1. Introduction

Double-strand break (DSB) repair response guards cells against lethal damage to DNA and is fundamental to cellular survival and genomic integrity [1,2,3,4]. DSBs arise in diverse biological processes such as DNA replication, meiotic recombination, and chromosome segregation. Many DSBs occur programmatically and are managed by a set of repair responses whose regulation plays an essential role in cell survival. Examples include DNA rearrangement in the immune response [5,6], programmed DNA elimination during chromosome maturation of protists and other eukaryotes [7,8], gene regulation [9,10], meiosis [11,12,13], yeast sex determination at the homothallic locus [14,15,16], and the response to mechanical stresses exerted on pericentric chromatin [17,18,19].

DSBs do not occur uniformly throughout the genome. The genomic locations of breaks change throughout the cycle of a single cell and are specific to different cell types. Such specialization requires precise coordination of a host of repair responses available to the cell. Repair responses preserve genomic integrity by employing steps from overlapping pathways. The local chromatin environment surrounding the break site influences pathway selection. Chromatin undergoes extensive modifications near the DSB site (histone modifications and variant substitutions) creating a dynamic scaffold for repair factor recruitment. Chromatin proximal to DSB sites exhibits increased mobility [20,21,22,23,24,25,26] and the broken ends are usually displaced to the nuclear periphery or the edges of chromosomal territories [22,27]. Mechanistic understanding of damaged DNA mobility can uncover the general principles behind fidelity and efficiency of repair and explain how disturbance of DNA repair mechanisms leads to alterations in DNA sequence, chromosomal aberrations, and cell death.

To understand the behavior of regions with DSBs within a heterogeneous environment, we employ a bead-spring polymer model. The model represents chromatin fibers as chains of interacting beads connected by nonlinear, finitely extensible, worm-like chain (WLC) springs, incorporating physical factors like chromatin stiffness, excluded volume effects, and chromosome confinement by the nuclear envelope and tethering sites [28]. New insights into the underlying properties of sub-nuclear compartments (e.g., the nucleolus, pericentric chromatin) arise by considering the addition and dynamics of crosslinking protein complexes (SMCs). Previously our group has shown that dynamic, protein-mediated chromosomal crosslinks play a crucial role in the self-organization of the nucleolus, i.e., the rDNA region enriched in condensin [28]. The kinetics of these crosslinks significantly influence genome-wide connectivity within the resulting domain, as exemplified by the optimized gene clustering and mixing of genes among clusters that emerge at specific timescales [29].

In the current study, we investigate chromatin dynamics following damage (a break in the spring between adjacent beads on one chromosome arm) in a region organized by crosslinking (ROX). In our mathematical model, we achieve this through implementing a break and adjusting the recruitment and behavior of crosslinking complexes at a specific locus in the ROX that encompasses beads near the break site. We observe that local depletion of the cross-links within the damaged region results in the region’s eviction from the cross-link-rich environment. Biologically, these genomic breaks occur in environments represented by chromosome territories (TADs), NOR (nucleolar organizer region), pericentric bottlebrush, or any region defined by the abundance and high activity of SMCs [17,19,28,30,31].

Experimentally, we demonstrate examples of pericentric chromatin displacement upon inducing damage to chromosome III of the budding yeast. Chromosome III had previously been modified to include a second conditional centromere (GalCEN) to induce double-strand breaks upon addition of glucose and a fluorescent reporter to track the position of LacO::LacI-GFP spot relative to the spindle upon damage. We observe increased displacement of pericentric chromatin (within ROX) following induced damage compared to damage introduced in the arm. The increase is dependent on the repair factor Rad52p, a well described regulator of DSB repair conserved throughout phylogeny. The dynamics of cross-link binding and dissociation within the ROX domain are key determinants in the magnitude of displacement and rates of partitioning of the damaged region.

## 2. Results

### 2.1. Determination and Visualization of the ROX Boundary

We define an ROX of interest in the genome, as described in the Methods section and as shown in Figure 1A,B. Previous work suggests that the presence of SMCs in a region of chromatin produces a phase-separated membrane-less nuclear sub-compartment, and thus our first goal was to effectively quantify and visualize the geometry of the ROX (relative to itself and the rest of the genome) over time. To do so, we defined the ROX boundary as the 3D surface that encloses 91–94% of all of the positions taken by ROX beads over the course of one simulation (Figure 1C, top); we refer to the volume contained within the ROX boundary as the ROX compartment. The 91–94% percentage range was chosen arbitrarily to be high enough for the ROX compartment to contain the majority of ROX bead positions throughout a simulation’s runtime, but still be able to detect expulsion events as beads exit the compartment. We generated a 2D experimental image by assigning a point-spread-function (PSF) to each of the beads (Figure 1C, bottom). The total signal is thresholded to delineate a boundary around the densest part of the region, computed by connecting regions where the total PSF value equals an arbitrarily chosen value such that the boundary encapsulates 91–94% of the point cloud. In Figure 1D, we show how to determine whether individual beads fall inside or outside the ROX boundary at a given point in time. Through tracking individual beads’ positions relative to the boundary, we can better understand the persistence of the ROX domain geometry and the degree to which ROX versus non-ROX beads explore space.

### 2.2. Depletion of Cross-Links Enables Eviction of the Damaged Region (20 Beads) from the ROX Compartment

Figure 2A depicts how different crosslinking conditions influence the locations relative to the ROX boundary of the 20 beads that compose the damaged region of the ROX. When all 361 beads in the ROX retain their dynamic crosslinking eligibility throughout the entire simulation, the damaged region primarily stays inside the ROX boundary, as shown by the light and dark blue boxplots. This is consistent over all observed crosslinking regimes, regardless of the introduction of a DSB; both rate of crosslinking and presence of a DSB have negligible effects on these beads’ locations relative to the boundary. Increasing the crosslinking rates from slow (10/1) to fast (0.1/0.01) reduces the spread of the distribution of bead distances, which supports the finding put forth in Walker et al. that the rigid clusters produced by the fast crosslinking regime constrain overall bead motion. In the simulations where the 20 beads composing the damaged region release their existing cross-links and lose the ability to form new ones (Figure 2A, light and dark red boxplots), we observed a dramatic exodus of beads from inside to outside the ROX boundary. That this trend persists across all observed crosslinking regimes highlights the key role of crosslinking in modulating the degree to which chromatin explores space. When crosslinking rates are kept consistent, introducing a DSB leads to a slight increase in median distance from the boundary. Similarly, faster crosslinking rates correspond with slightly increased average distances traveled by beads beyond the outside of the boundary. We note that the definition of the “ROX” boundary and corresponding statistics are less meaningful in the no crosslinking simulations, as (other than the potential existence of a DSB between beads 240 and 241) there are no notably different dynamics distinguishing the 20 beads in the damaged region from the rest of the beads composing the ROX, nor distinguishing the beads composing the ROX from the rest of the genome.

We see the same overall trends from Figure 2A reflected in Figure 2B. Whereas Figure 2A provides a sense of the degree/distance to which beads from the damaged region move outside or remain inside the ROX boundary, Figure 2B shows the percentage of time beads spend outside the ROX boundary. In the cases involving local crosslink depletion, the beads composing the damaged region spend more than 50% of the simulation time outside the ROX boundary; median percentages increase by ~20% as we tune crosslinking rates from slow to fast. In comparison, in simulations involving persistent crosslinking eligibility, beads in the damaged region venture outside the ROX boundary relatively rarely. The exclusion of beads via cross-link removal is a powerful first step in conferring a means of genome searching to a specific DNA segment.

### 2.3. Depletion of Crosslinks Proximal to the Break Increases the Area of Exploration of Affected Region

In order to better understand the extent to which damage affects the spatial confinement versus mobility of nearby chromatin, we computed the radii of confinement of beads within and beyond the damaged region, as described in the Materials and Methods section. As shown in Figure 3, this ~200 kb region centers on the DSB location between beads 240 and 241 and extends 100 kb past each end of the damaged region. We observe that faster crosslinking kinetics result in stronger confinement of the individual beads compared to intermediate and slow kinetics.

This result is in agreement with previous trends reported in [29] where rigid and flexible clustering were shown to emerge for different crosslinking regimes. Comparing Figure 3A,B, we note that it is the presence and rate of crosslinking rather than the existence of a DSB that dictate the motion of the beads. Although a modest increase in average radii of confinement of beads immediately proximal to the break site occurs in the slow (10/1) crosslinking regime in response to a DSB event, this effect is quite localized and disappears at faster crosslinking regimes.

In contrast, locally depleting cross-links from the 10 beads (beads 231 to 250, highlighted by the shaded region in Figure 3) on either side of the break site dramatically increases the radii of confinement of those beads. Figure 3C,D show that this increase (to a radius above ~750 nm) is quantitatively equivalent across all observed dynamic crosslinking rates and for all beads in the damaged region, and occurs regardless of the existence of a DSB. Figure 3D additionally highlights the cumulative effect of both local cross-link depletion and a DSB on bead motion, as beads’ average radii of confinement increase to or above ~1000 nm as we move along the damaged ends of the chromosome arm towards the break site. When viewed alongside the results shown in Figure 2, this work suggests that, in addition to their eviction from the ROX compartment, ROX beads experiencing local depletion of crosslinks exhibit up to a 3-fold increase in mobility, depending on the initial binding kinetics of the removed cross-links. Interestingly, we observe that the average radii of confinement of the remaining beads in the ROX (i.e., those not part of the damaged region) remain approximately the same (Figure 3A–D), *regardless of the rate of crosslinking present in the undamaged region or the type/extent of damage in the damaged region*. This suggests that local cross-link depletion is an effective method through which selected portions of the genome can be targeted for increased mobility while still preserving preexisting average confinement levels and crosslinking dynamics elsewhere.

### 2.4. Local Depletion of Cross-Links Amplifies Distance Between Damaged Ends Following a DSB, Regardless of Crosslinking Rate Elsewhere in the ROX

We then narrowed our focus to the behavior relative to each other of the two beads on either side of the break site; namely, we computed the pairwise distances between beads 240 and 241 over time and compared results among simulations involving different crosslinking and damage conditions. In simulations involving an intact chain (no break), we observed that locally depleting (i.e., “releasing”) cross-links had little effect on the average distance between beads 240 and 241. As seen in Figure 4A–C, these average distances oscillate between 15 and 45 nm over time and are also largely independent of the rate of dynamic crosslinking present. This suggests that, though local depletion of crosslinks enables beads 240 and 241 to experience increased average radii of confinement values, the average distance between the two beads is not sensitive to depletion as long as there is no DSB present. We note that the standard deviations (red and blue shading) shown in Figure 4 reflect the fact that beads 240 and 241 may have different pairwise distances due to random noise at the single cell level, as well as the potential to belong to different clusters over time and across different random seeds. Though plotting data averaged over 10 runs as we do in Figure 4 therefore obscures some run-specific behavior, particularly with regard to illuminating the timing and frequency of bead movement to new cluster(s) in the fast crosslinking regime, the corresponding single cell analysis does not contradict the qualitative conclusions drawn here.

The introduction of a DSB has the potential for a marked effect on average pairwise distance between beads 240 and 241, depending on crosslinking dynamics and presence in the ROX (Figure 4D–F). In virtual experiments where all ROX beads remain eligible for dynamic crosslinking throughout the entire simulation, the average distance between the two beads proximal to the DSB depends on the rate of crosslinking present in the ROX. More specifically, slower crosslinking regimes allow for increased average distances following a DSB (Figure 4D,E), while faster crosslinking regimes do not (Figure 4F). In the fast crosslinking regime, average distance following a DSB remains qualitatively similar to that seen in the no break case, with a comparison of the blue trends in Figure 4C,F actually showing small reduction in pairwise distance for the DSB case. Considered in the context of prior work linking crosslinking rate and chromatin substructure [28], this outcome suggests that the clusters present in the fast crosslinking regime continue to constrain the movement of the end beads following a DSB. Interestingly, when the damaged region of the ROX experiences local crosslink depletion in addition to a DSB, the average distances between beads 240 and 241 increase beyond the values observed in the virtual experiments with only a DSB and no local depletion. This increased distance between the end beads occurs rapidly following the damage events at the 2500 timestep and, crucially, regardless of the rate of crosslinking present in the undamaged region of the ROX. Our simulations show that, across all observed crosslinking regimes (red trends in Figure 4D–F), the end beads reach separation distances of over a micron within the first 500 s.

Overall, we conclude that the cumulative effect of local depletion and a DSB produces the largest pairwise distances between beads 240 and 241. That this outcome occurs regardless of the rate of crosslinking present in the undamaged portion of the ROX suggests a potential mechanism through which broken chromatin ends can reliably move apart, even if originally part of a region of enhanced connectivity and compaction prior to the damage event. These results suggest that crosslinking spatiotemporal dynamics in the ROX can regulate the motion of the damaged ends proximal to a DSB, thus influencing DNA repair, including both homology- and non-homology-based pathways.

### 2.5. Depletion of Cross-Links Has Primarily Local Rather than Global Effects

Figure 5 further exemplifies the segregation effects of locally depleting cross-links. Here, we use contact maps to visualize the 3D Euclidean pairwise distances between beads in the ROX over both time and rates of crosslinking. As demonstrated previously, the rate of crosslinking significantly influences bead-to-bead distances, with faster rates corresponding to smaller distances among beads in the ROX and slower rates corresponding to generally larger/more varied distances and more uniformity throughout the ROX. This pattern is particularly noticeable in the case where no dynamic binding is present. We show that depletion of cross-links has primarily local rather than global effects, as the pairwise distances among those beads in the ROX that are still able to form cross-links remain relatively similar over time, in contrast with the increased pairwise distances between these beads and the 20 beads in the “depleted zone” that have lost the ability to form cross-links. This distinction becomes more pronounced over time, though the effects are noticeable soon after the depletion event occurs (Appendix A). Interestingly, the pairwise distances between beads in the depleted zone and other beads in the depleted zone are not as large as those between beads in the depleted zone and beads outside the depleted zone. This is particularly noticeable in the case of fast crosslinking, in which pairwise distances between beads in the depleted zone with other beads in the depleted zone appear to initially increase before decreasing in some cases to previous levels. This suggests a physical segregation of the ROX into two regions: beads that are still able to dynamically cross-link versus beads that are not.

### 2.6. Pericentric Chromatin Is Displaced from the Bottlebrush upon DNA Damage

Conditional dicentric yeasts are made by insertion of an additional centromere under a control of a Gal1-10 promoter (GalCEN) at loci with approximate coordinates (at ~HIS4 65,000 bp; GBP2 102,000 bp; ILV6 105,000 bp) upstream of endogenous centromere on chromosome III (for details refer to [32]) into a strain with a FROS (10 kb LacO array) inserted 3.8 kb downstream (at 118,500 bp) of the centromere. This arrangement places GalCEN in the arm (HIS4) and the LacO as well as the conditional centromere (GalCEN at GBP2 and ILV6) into pericentric chromatin, region of SMC complex enrichment [33,34] and as defined by Paldi et al. [35] for *S. cerevisiae*. The cartoon in Figure 6 depicts relative locations of the insertions with numbers in blue boxes indicating distance (in kb) away from the endogenous centromere.

Upon addition of glucose to the growth media, galactose promoter is repressed and the dicentric chromosome is activated. Both functional centromeres (endogenous CEN3 and GalCEN) recruit kinetochores and about 50% are pulled toward opposing spindle poles, thus creating a physical conflict which results in a chromosome break. Most DSBs accumulate near the centromeres, with a small fraction occurring in between the two centromeres [17,36]. When the GalCEN is placed 46.3 kb away from CEN3 we observe a pericentric LacO, positioned near CEN3 (Figure 6) displaced from the spindle axis (SPB-SPB) in only 3% of cells. In strains with GalCEN positioned 12.3 kb and 9.8 kb from CEN3 (at GBP2 and ILV6, respectively) the dicentric chromosome is predicted to break in the pericentrome. However, following the DSB event in those strains, the LacO is displaced from the spindle in 21–22% of observed cells (*p*-value < 0.0001; Appendix A). This is evidence of a high degree of re-organization of pericentric chromatin (also known as the bottlebrush) following a local break. Pericentric chromatin of a mitotic chromosome is typically enriched in loops stabilized by an accumulation of crosslinking proteins (SMCs). The increase in LacO displacement following a break in the centromere suggests that cross-linking is extensively and transiently diminished at regions proximal to the break as a step of a molecular pathway preceding efficient repair.

In pericentric GalCEN strains (at GBP2 and ILV6) which lack an essential homologous repair factor, Rad52, the difference in LacO displacement is reduced to 1–5% (Figure 6; see Appendix A for comparison statistics). We consider displacement to be when the centroid of LacO reached above 0.4 µm away from the SPB-SPB axis and do not quantify displacement differences between different strains. Select images of displaced LacO spots are shown in Figure 6c,d.

## 3. Discussion

Packaging millimeters of DNA on the order of several thousand-fold into the size of a micron-scale nucleus constrains access to DNA for processes such as repair and transcription. Despite the immense spatial compaction and confinement of DNA, specific genetic loci are able to be identified for transcriptional activation or deactivation. In addition, specific loci are used as templates for DNA repair of homologous loci that have experienced DNA damage (e.g., DNA double strand breaks) [1]. The results provided herein provide insights into the search mechanisms that underlie the ability of different regions of the genome to be identified. Chromosomes occupy individual territories, reflecting the propensity of individual polymer chains to adopt random coil configurations that maximize entropy [37,38,39,40]. In addition to the behavior of the chain, a driving principle for chromosome organization is the distribution of cross-links within a chain and between chains [28,29]. A class of structural maintenance of chromosome (SMC) proteins exhibit the ability to crosslink DNA chains [41,42,43]. Through a combination of computational modeling and experimental results, we find that regulation of distribution and affinity of crosslinks is a major driver of chain motion and structure [44].

Crosslinking can be modeled by adding additional springs between pairs of beads that stochastically form (when they are within a prescribed distance) and break according to a timescale t_on_, which dictates the mean duration of a transient bond [28,29]. The presence or absence of these cross-links, and whether they are dynamic or fixed, results in different configurations and material properties of sub-domains within the nucleus. The most pronounced nuclear sub-domains are the nucleolus and pericentromere that house the ribosomal DNA and centromere satellite DNA, respectively [34,45,46]. In the nucleolus, the cross-linkers promote gene clustering, most likely representing regions of transcriptional activity. In the pericentromere, the cross-linkers facilitate a dense array of loops, known in polymer physics as a bottlebrush, which generates tension necessary for faithful chromosome segregation. These emergent behaviors are two examples of higher order structures arising from crosslinking that specify discrete functionalization of the genome [34,47].

Depletion of cross-links is another way to functionalize a region of the genome for a specific process. In this instance, the process is the DNA damage response. Experimental studies suggest that cells rapidly modify the chromatin surrounding the site of DNA damage [48,49,50,51,52]. The study herein predicts that damage modifications include direct effects on the binding kinetics of cross-links as well as indirect effects through histone modifications that impinge on crosslinking kinetics. Reductions in cross-links result in the near-immediate expulsion of a DNA locus from the interior of a pre-established chromosome territory. Exposure of the damaged locus to the nucleoplasmic milieu will enhance encounters with repair enzymes for non-homologous end-joining and/or regions of homology for recombinational repair. The rapid response between loss of chromatin cross-links and release from chromosomal territory constraints (Figure 4) heightens the biological relevance.

The direct demonstration that DNA is expelled from compartmentalization in vivo was from studies of sites of DNA damage in the nucleolus [31]. In this situation, the damaged site traveled away from the nucleolus where it was able to interact with the Rad52 recombinational repair machinery in the nucleoplasm. The phenomenon was rationalized as a mechanism to prevent recombination of ribosomal DNA repeats, which would have the negative consequences of shuffling the rDNA content. In addition, our work demonstrates that migration of damaged locus from regions of chromosome territories pre-established by crosslinking may be a general mechanism for effective DNA repair. In order to repair DNA damage, the local chromosome structure must be reorganized into a permissive material that allows recruitment of repair factors. Chromatin remodeling is known to play an integral role in DNA repair and damage pathway choice [53,54,55]. Dissolution of cross-links may be the first step toward satisfying this requirement.

To test the generality of expulsion from regions of crosslinking we experimentally induced breaks in a region enriched in chromatin cross-links, the pericentromere. The pericentromere is enriched three-fold in condensin and cohesin, but unlike the nucleolus, the pericentromere of budding yeast is composed of unique DNA sequences. Upon induction of a DNA break via activation of a dicentric chromosome, the damaged DNA is relocated outside the pericentromeric compartment that is enriched in DNA cross-links (Figure 6). More than one fifth (22%) of the damaged DNA is released, indicative of a general mechanism to free sites of DNA damage from the confines of their local environment. The finding that damaged DNA from the yeast pericentromere behaves like damaged nucleolar DNA [31] indicates that it is not sequestration of rDNA repeats in the nucleolus, rather it is physical translocation of mobilized DNA ends enabled by the release of confining cross-links. The lower fraction of DNA that migrates away from the pericentromere (22%) relative to the 50% of the damaged DNA that relocates from the nucleolus [31] reflect differences in the extent of DNA damage (fraction of molecules damaged). Damage within the nucleolus was introduced via homing endonucleases that cut close to 100% of the molecules. Damage in the pericentromere, presented herein, was induced via dicentric chromosome breakage that cuts 50% of molecules due to the polarity of attachment of sister chromatids in mitosis.

In other experimental studies on ROX (regions organized by cross-linking) in budding yeast, a reduction in cross-links within the rDNA results in changes to the visco-elastic property of the compartment and its reorganization [56,57], thereby exerting control over the fate of DNA associated proteins. The relocation to the compartment’s periphery of Cdc14, an rDNA phosphatase involved in segregation and required for mitosis, is observed as a consequence of reduced crosslinking [58]. These examples illustrate how the abundance of cross-links or their dynamic activity can have biological effects on density and the site of activity of chromatin-associated factors and thus provide local feedback toward their function in DNA damage.

The modulation of cross-links at the site of break as a general mechanism for facilitating DNA repair has important physical implications. Escape from the interior of the chromosome territory is a critical first step in freeing the damage site to explore the nucleoplasm for repair machinery, or ectopic chromosomal domains for homology. That damage within the nucleolus as well as the pericentromere lead to experimentally observed translocation of damaged DNA suggests that this may be a universal mechanism. While there may be active mechanisms that drive DNA motion, the release of cross-links is sufficient to expel DNA from the chromosome territory and increase its range of motion. This study highlights another powerful organizational feature that emerges through the use of cross-links as chromosome building blocks.

## 4. Materials and Methods

### 4.1. Experimental Materials and Methods

Budding yeast cells were transformed with a GALCEN3 fragment directed to three genomic positions on chromosome III. These are ILV6 (9.8 kb from CEN3), GBP2 (12.3 kb from CEN3) and HIS4 (46.3 kb from CEN3). Growth on galactose as the sole carbon source inactivates the GALCEN3 and allows maintenance of the conditional dicentric chromosome in its monocentric state. Upon switching to glucose as a source of carbon the conditional centromere (GALCEN3) is activated and the dicentric chromosome will eventually break in proximity to either of the centromeres. To visualize the motion of chromatin near the chromosome break, a 10 kb array of *E. coli* lac operator DNA was introduced 3.8 kb from CEN3 on the right arm of chromosome III. The operator DNA was visualized with a lac repressor-GFP fusion protein integrated at the URA3 locus. All strains were transformed with a spindle pole component (Spc42) fused to mCherry in order to visualize the chromatin::LacO spots relative to the mitotic spindle in living cells. The RAD52 gene was removed in selected strains by gene replacement with LEU2. The seven strains used in this study are listed in the strain table.

For analysis of DNA damage, cells were switched from galactose to glucose YPD media (1% Yeast extract, 2% Bacto-peptone, 2% Dextrose supplemented with excess adenine) for 3 h. Growth on glucose results in activation of the dicentric chromosome and DNA damage after 2–3 h [17]. Strains were grown until mid-logarithmic phase prior to imaging. Images were acquired at room temperature using a Ti-Eclipse inverted microscope (Nikon, Melville, NY, USA) with a 100 × Plan Apo 1.4 NA objective (Nikon, Melville, NY, USA) and Clara CCD digital camera (Andor Technology, Concord, MA, USA) using MetaMorph 7.7 imaging software (Molecular Devices, Downington, PA, USA). Single stacks contained seven Z-plane sections with 300 nm step-size. Image stacks were cropped, and maximum intensity projections were created using ImageJ for analysis (ImageJ 2.9.0, NIH, Bethesda, MD, USA).

### 4.2. List of Strains: Yeast Strains Numbers (Genotypes) Used in Experiments


**Strain ID:**


**KBY6450.1** (DCY1327.1) (CEN3(3.8)-GFP MATa ade2 his3 trp1 ura3 leu2 can1 LacINLSGFP:HIS3 lacO::URA3 (at 3.8 kb from CEN3, 10 kb array)) + spc42-mCherry:nat GALCEN3:Hb (40 kb) isolate 1

**KBY6561.1** (KBY6450) (CEN3(3.8)-GFP MATa ade2 his3 trp1 ura3 leu2 can1 LacINLSGFP:HIS3 lacO::URA3 (at 3.8 kb from CEN3, 10 kb array)) + spc42-mCherry:nat GALCEN3:Hb (40 kb) Rad52-CFP:KAN isolate 1

**KBY6622.3** (DCY1327.1) (CEN3(3.8)-GFP MATa ade2 his3 trp1 ura3 leu2 can1 LacINLSGFP:HIS3 lacO::URA3(at 3.8 kb from CEN3, 10 kb array)) + spc42-mCherry:nat GALCEN3:Gbp2:Hb isolate 4

**KBY6625.3** (KBY6622.3) (CEN3(3.8)-GFP MATa ade2 his3 trp1 ura3 leu2 can1 LacINLSGFP:HIS3 lacO::URA3(at 3.8 kb from CEN3, 10 kb array)) + spc42-mCherry:nat GALCEN3:Gbp2:Hb rad52∆Leu2 isolate 4

**KBY6624.1** (KBY6622.3) (CEN3(3.8)-GFP MATa ade2 his3 trp1 ura3 leu2 can1 LacINLSGFP:HIS3 lacO::URA3(at 3.8 kb from CEN3, 10 kb array)) + spc42-mCherry:nat GALCEN3:Gbp2:Hb Rad52-CFP:KAN isolate 4

**KBY6449.3** (DCY1327.1) (CEN3(3.8)-GFP MATa ade2 his3 trp1 ura3 leu2 can1 LacINLSGFP:HIS3 lacO::URA3 (at 3.8 kb from CEN3, 10 kb array)) + spc42-mCherry:nat ILV6:GALCEN:Hb isolate 3

**KBY6497.2** (KBY6449.1) (CEN3(3.8)-GFP MATa ade2 his3 trp1 ura3 leu2 can1 LacINLSGFP:HIS3 lacO::URA3 (at 3.8 kb from CEN3, 10 kb array)) + spc42-mCherry:nat ILV6:GALCEN:Hb rad52DLeu2 isolate 2

### 4.3. Simulation Parameters and Data Collection

We use a bead-spring polymer chain model to simulate the spatiotemporal dynamics of chromatin of haploid budding yeast. Through discretizing each chromosome arm into non-overlapping 5 kb segments, we represent the yeast genome in silico as 32 doubly tethered chains of beads connected by worm-like chain (WLC) springs. For a detailed discussion of baseline model parameters and forces we refer the reader to [37]. In our model, we designate a contiguous segment (1.8 Mb, 361 beads) on the largest chromosome (right arm of chromosome XII; 2.7 Mb, 546 beads) as a region organized by cross-linking (ROX). Biologically, this segment is analogous to the rDNA locus in yeast. The beads in this ROX are subject to the same forces as all beads in the system; however, they differ in that they are also eligible to form transient pairwise bonds (i.e., cross-links) with other beads in the ROX. These transient or “dynamic” cross-links serve as proxies for SMCs (cohesin, condensin). We specify the binding rates and densities of the transient crosslinks in the model simulation parameters. As a central goal of this work is to study the effect of damage in a region of SMC protein-mediated crosslinking, we create such a region by making all ROX beads eligible for dynamic crosslinking (unless otherwise specified) and thus able to stochastically switch between “active” and “inactive” states over time. Each eligible bead switches from “active” (able to dynamically bind with another active bead) to “inactive” (unable to form/maintain a dynamic bond with another bead) following a duration drawn from N(μ, (μ/5)^2^); similarly, a bead may switch from “inactive” to “active” following a duration drawn from N(μ/10, (μ/50)^2^). In this work, we focus primarily on SMC-protein crosslinking timescales of μ = {0.1, 0.19, 0.3, 1, 10} seconds. For a thorough explanation of crosslinking implementation please see [28,29]. We simulate DNA damage through two approaches integrated into our computational model:(i)Double-strand break in the chromosome. In the first approach, we cut or “break” a chromosome arm at a designated point in time by permanently removing the WLC spring that connects two specific neighboring beads. Note that we refer here to the standard, entropic spring force that exists between neighboring beads on the bead-spring chain, not the stronger transient spring force that corresponds to a dynamic crosslink between two beads. We choose a break site near the center of the ROX, as seen in Figure 1B. Beads 61 through 421 of the right arm of chromosome XII (361 beads in total) correspond to the ROX (Figure 1A), and the break occurs between beads 240 and 241 (i.e., ~1.2 Mbps from CEN12).(ii)Local depletion of SMC proteins. In our second approach, we depleted SMC proteins (protein-mediated cross-links) near the break site. We simulate this depletion in the model by permanently removing selected beads in the ROX from the pool of beads that are able to dynamically cross-link. More specifically, we require that, from a specific point in time onwards, the 20 beads representing the damaged region of the ROX (i.e., the 10 beads on each side of the break site, corresponding to ~50 Kb each, based on the distribution of modified histone (isoform H2AX) at sites of DNA damage [59]) release their cross-links and are no longer eligible to form dynamic cross-links. We set the time of this depletion to be 2500 s after the start of the simulation, i.e., at the same time as the DSB.

In Table 1, we highlight the datasets generated using our mathematical model and used in this work. In Appendix A, we summarize relevant timescale choices and parameters.

### 4.4. Distance to Edge of ROX Domain

To determine the ROX domain boundary, we first discretize the 3D nuclear space into a grid of non-overlapping cubic cells of side length 20 nm. We then apply a Gaussian point-spread function (PSF) at each set of xyz-coordinates of the ROX beads post-break and assign to each cell the total PSF at its location. Finally, the center of each cell with a total PSF approximating 300 is used as a vertex of a 3D surface outlining the ROX domain. The PSF threshold value is chosen empirically such that the resulting surface encapsulates the region organized by crosslinking (ROX). To ensure that the ROX boundary lies within the confines of the nucleus, any vertices outside of the nucleus are projected onto its surface. After the vertices have been established, the xyz-coordinates of the bead of interest are used to compute the distance to the closest vertex at each timepoint. Because the number of vertices is large (~10 K) and their distribution on the nucleolar boundary is uniform, the distance between these two points is a reasonable approximation for the distance from the bead of interest to the boundary. To determine whether the bead is located inside or outside of the ROX at every timepoint, we use the intensity of the point cloud at the bead’s location. Intensity greater than the threshold used for generating the surface indicates that the bead is within the ROX, whereas a smaller intensity indicates the bead lies outside the ROX domain. We include a description of an alternative “geometric” method for determining the bead’s position relative to the ROX boundary in Appendix A. The results captured in Figure 2A and Appendix A computed using “point-spread function” and “geometric” methods, respectively, are in accordance and analogously capture the differences between crosslinking regimes.

### 4.5. Radius of Confinement

We use radii of confinement to quantify the mobility of beads in the damaged region by adapting the equation from [60] to our 3D data:(1)$$R_{c}=5/6*sqrt(6\sigma^{2})$$
where $R_{c}$ is radius of confinement and *σ* is a bead’s standard deviation from its average position over the course of single simulation. We compute the averaged radii of confinement with standard deviations for each bead in the ROX from (N = 10) simulations grouped by the condition (break vs. no break) and the crosslinking regime.

### 4.6. Distance Between Cut Ends

In order to better understand the localized effect of DNA damage, we first computed the 3D Euclidean distance between the two beads located on either side of the break site. We computed this pairwise distance at each step over time and then averaged over multiple runs.

### 4.7. Point Cloud Overlap

To quantify the degree to which the damaged region is expelled from the region organized by cross-linking (ROX), we quantify the overlap between the point cloud of positions taken by beads in the damaged region over time (point cloud A) and the point cloud of positions taken up by other ROX beads (point cloud B). We measure overlap as the number of beads in point cloud A that are within 40 nm of at least one point from point cloud B, which we report as a percentage of the total number of points in point cloud A. See Appendix A.

## 5. Conclusions

A consequence of the territorial organization of the genome is the tendency for DNA to be confined within the body of the chromosome. We have used a polymer bead-spring model of a dynamically crosslinked chromosome to model how a broken DNA end escapes the confines of the chromosome. Specifically, we show that removal of chromosome cross-links is sufficient to release the damaged DNA end from the chromosome body. The dynamic crosslinking regimes govern the kinetic exploration of genomic space by the damaged region and control the distance between its ends. We provide an experimental example with live-cell microscopy by tracking pericentric regions of budding yeast chromosomes following dicentric chromosome breakage. The broken DNA end is released from the pericentromere region and explores space beyond the zone of confinement. Our work highlights the role dynamic crosslinking by SMCs potentially plays in homologous recombination and/or non-homologous end-joining repair pathways following DNA breaks.

## Figures and Tables

**Figure 1 ijms-26-11697-f001:**
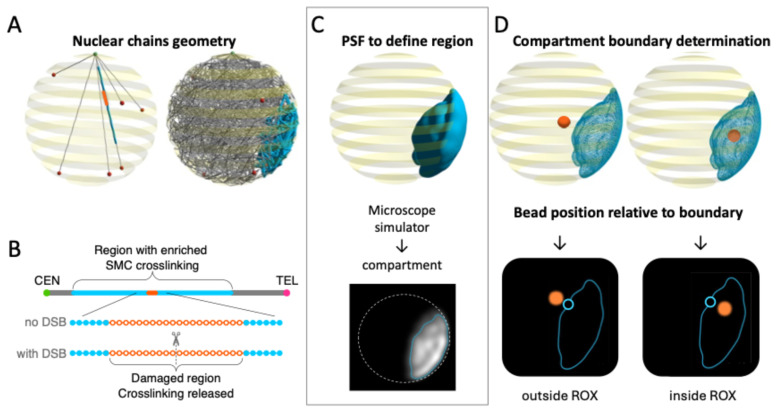
Simulation design and analysis pipeline. (**A**) Snapshots of the simulated haploid yeast chromatin genome before (left) and after (right) equilibration. The 32 bead-spring chains, analogous to the 32 arms composing the 16 yeast chromosomes, are tethered to the spindle pole body on the nuclear membrane through the centromere. The 32 telomeres (red) are tethered to 6 sites on the nuclear membrane distal from the centromeres (green). The region of enriched crosslinking (ROX) is denoted by blue and orange beads. Left: Initial configuration of overlapping 32 chromatin chains. Right: Configuration of 32 chromatin chains after dynamic equilibrium (2500 s of simulation run). (**B**) Schematic the region of enriched crosslinking (ROX). Beads eligible for dynamic crosslinking are indicated in blue filled circles; beads that lose the ability to form dynamic cross-links following a DSB event (i.e., the “region of damage”) are indicated as orange open circles. All other beads are shown in gray. (**C**) The top image shows the surface of the ROX. The bottom image is a slice through the simulation as it would appear in a biological specimen viewed through a microscope objective lens. PSF (point spread function), used to extract ROX boundary, refers to a mathematical transformation depicting how light would spread through the microscope’s viewing objective based on bead’s position (see Section 4: Materials and Methods). (**D**) Depiction of two beads of interest (orange) relative to the ROX domain surface. Top: Simulation snapshots with the surface area of the ROX domain and surface beads are shown as a mesh. Bottom row: The bottom image is a slice through the simulation showing the contour of the ROX and the two beads (blue is on the surface of ROX; orange is outside the region). These panels illustrate how the region of chromatin crosslinking is determined, and how the DSB with local cross-link depletion is mobilized away from the ROX domain.

**Figure 2 ijms-26-11697-f002:**
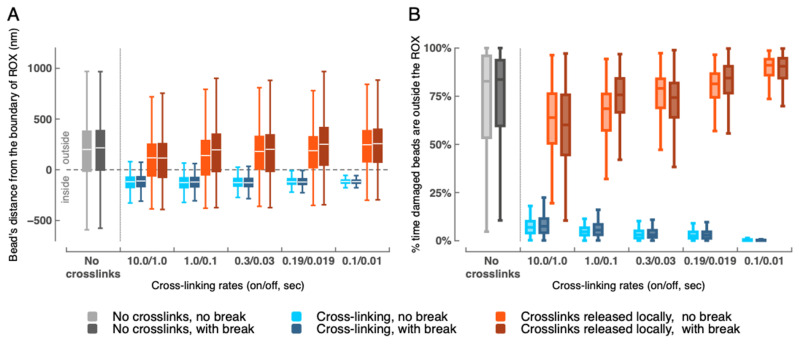
Ability to form cross-links determines bead spatial location relative to the ROX boundary. (**A**) Boxplots depict the distributions of distances of beads within the damaged region relative to the ROX boundary from 2400 to 3700 s after the start of the simulation across different crosslinking regimes. More specifically, each datapoint in a given boxplot corresponds to the distance from bead_i_ (where bead_i_ is one of the 20 beads in the damaged region) to the ROX boundary at a single point in time for a single simulation run. Each boxplot therefore considers 260,000 total datapoints (20 bead-to-ROX-boundary distances × 1300 timesteps × 10 runs). Negative distances and positive distances refer to beads located inside and outside the ROX boundary, respectively. Note that while the boxes bound the IQRs (interquartile ranges), the central white lines indicate mean (not median) values. Simulations involving “No cross-links” are included as controls; although there is no crosslinking present, we use the same methods to compute the “ROX” compartment and determine the relative damaged-region-bead distances as in all other cases. (**B**) Percentage of time that the beads composing the damaged region are found outside the ROX boundary, computed over the same 1300 s window. Each boxplot therefore considers 200 total datapoints (percentages of total time for each of the 20 beads × 10 runs). These boxplots show the medians (not means) and IQRs for the distributions of these percentages across dynamic crosslinking regimes.

**Figure 3 ijms-26-11697-f003:**
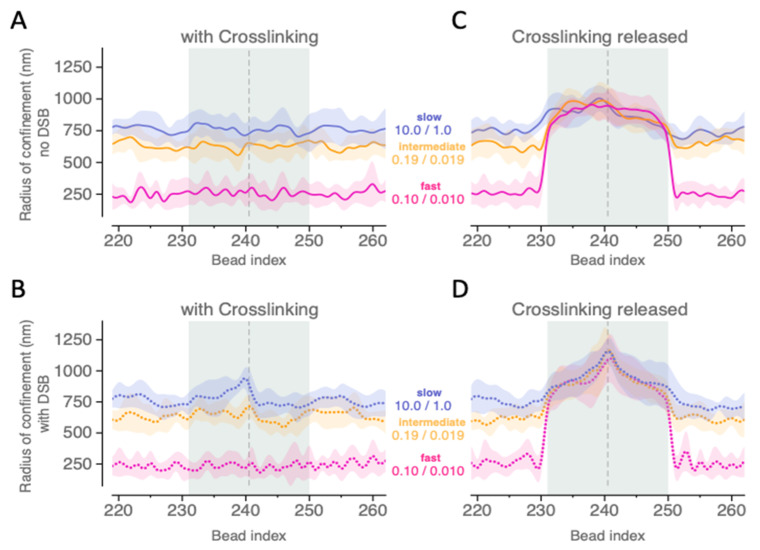
Radius of confinement of beads in the region near the break site. The break site (vertical dashed line) corresponding to a potential DSB event is located between beads 240 and 241. In panels (**A**–**D**), we show the local effects on radii of confinement of the 21 beads on each side of the break site (beads 220 to 261) over three rates of crosslinking: “slow” (blue), “intermediate” (yellow), and “fast” (pink). We compute radii of confinement and associated standard deviations (trend lines and matching shaded regions) for individual beads, averaging over time (1300 total seconds) and random noise (N = 10 runs). Solid lines in panels (**A**,**C**) show trends absent the break; dotted lines in panels (**B**,**D**) show trends with the break. Local depletion of cross-links on both sides of a break site allows for increased motion of the beads composing the damaged region of the ROX (highlighted here in gray, see Figure 1B for specifics).

**Figure 4 ijms-26-11697-f004:**
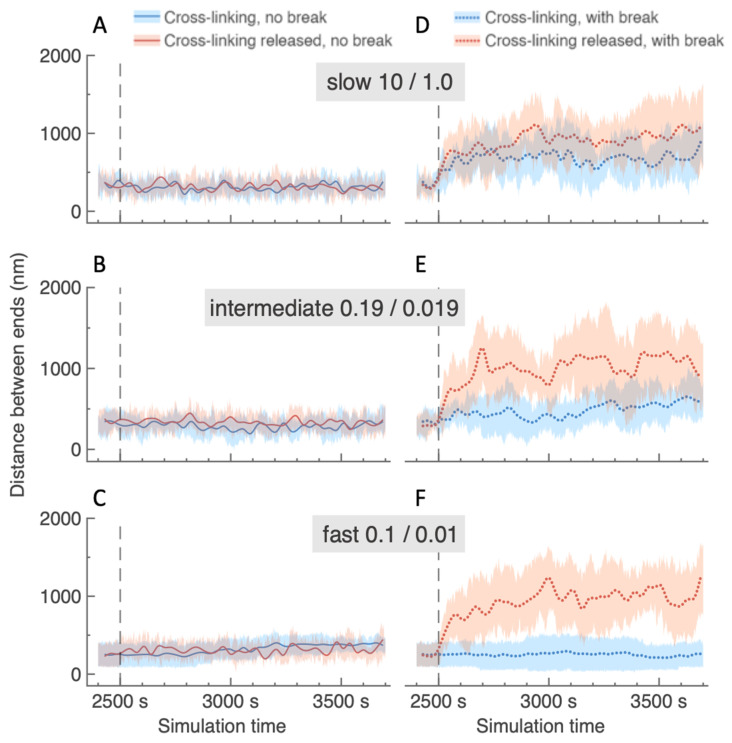
Distances between the two beads immediately proximal to the break site. Plots (**A**–**F**) show 3D Euclidean distances between beads 240 and 241 over time, with each row corresponding to a different crosslinking regime and each column corresponding to a different DSB setting. Crosslinking regimes are indicated in the gray boxes above (slow 10/1.0 t_on_ s/t_off_ s, intermediate 0.19/0.019 t_on_ s/t_off_ s, fast 0.1/0.01 t_on_ s/t_off_ s). As before, the DSB (vertical dashed line) between beads 240 and 241 occurs at timestep 2500 s. Each plot (**A**–**F**) shows average distances (with standard deviation) of 10 randomly seeded experiments where ROX beads in the damaged region: (i) remain eligible for dynamic crosslinking throughout the simulation (blue), or (ii) release any current cross-links and lose their eligibility to form new ones at the time of the break (red). Only following a DSB (panels (**D**–**F**)) do the ends show significant separation, particularly at faster binding kinetics.

**Figure 5 ijms-26-11697-f005:**
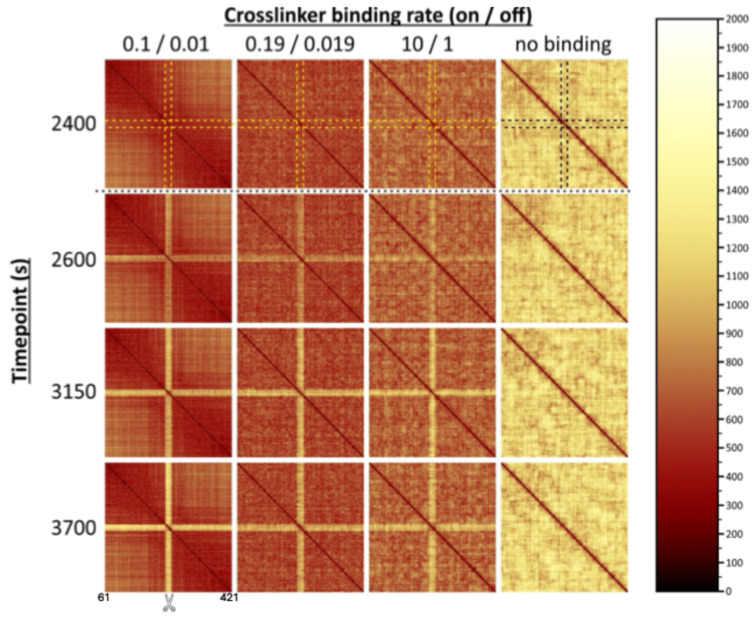
Proximity maps depicting 3D bead-to-bead distances within the ROX at selected points in time. Proximity maps reveal the degree of bead separation (bright yellow cross pattern over time) following depletion of cross-links from the damaged region relative to the rest of the ROX. Each proximity map displays data corresponding to the ROX (361 beads from 61 to 421 of the right arm of Chromosome XII as depicted in Figure 1) and was produced through averaging bead-to-bead distances for 10 distinct simulations at a specific point in time. The 16 plots composing this figure consider 40 total simulations, all reflecting an intact ROX with localized cross-link depletion. The four plots in a given column correspond to the rate of crosslinking in the ROX (“fast” 0.1/0.01 s; “intermediate” 0.19/0.019 s; “slow” 10/1 s; “no crosslinking”) across, at 2400, 2600, 3150, 3700 s of simulation, from top to bottom. Beads closer to each other are indicated by darker colors, bead separation is depicted in lighter colors (scale on right). Depletion of cross-links begins 2500 s after the start of simulation and is shown by the dotted line. Position of the chromosomal break (between beads 240 and 241) is marked by scissors. Cross-links are depleted from beads 231–250, indicated by the dashed yellow lines in the t = 2400 s (top row, prior to depletion. Beads 231–250 remain in proximity prior to 2500 s. Separation of the damaged region is visible at 100 s after depletion (seen in 2600 s row as a yellow cross in the center of the proximity map, growing brighter as time progresses (2600–3700, going down). Color bar units are in nm.

**Figure 6 ijms-26-11697-f006:**
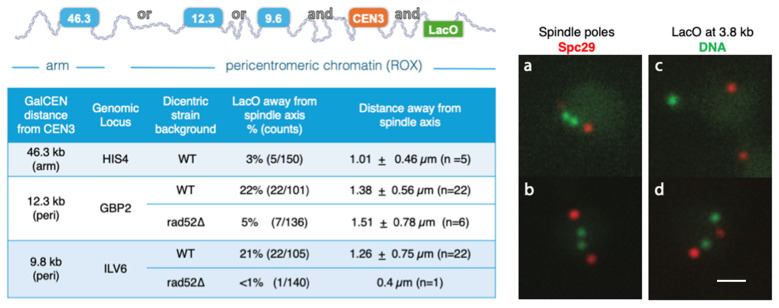
Movement of pericentric LacO away from the spindle following DNA breaks. Cartoon depicting chromosome III with locations of the second conditional (GalCEN) centromere in dicentric strains relative to endogenous (CEN3) centromere. Displacement of the LacO occurs about 20% of the time in the strains where the activated GalCEN is located in the pericentromere of chromosome III, and <5% of the time when the GalCEN is the arm (46.3 kb). (**a**–**d**) Examples of spots in the dicentric yeast strains with GalCEN at 9.8 kb. Two LacO spots are on the spindle axis in (**a**,**b**). In (**c**), a single LacO spot is observed away from the spindle axis. In (**d**), a single LacO spot is on the spindle axis and a second spot is a short distance away. LacO DNA is visualized with LacI-GFP (green spots). Spindle pole bodies are visualized with Spc29-RFP (red spots). Scale bar is 1 micron.

**Table 1 ijms-26-11697-t001:** Overview of simulated data generated and used in analyses. Sets of 10 runs differ based on crosslinking rate, existence of a break, and/or local cross-link depletion.

Crosslinking Rate (on/off) in s	Break Induced	Cross-Links Released Locally	Number of Independent Runs
0.1/0.01	no	no	10
		yes	10
	yes	no	10
		yes	10
0.19/0.019	no	no	10
		yes	10
	yes	no	10
		yes	10
0.3/0.03	no	no	10
		yes	10
	yes	no	10
		yes	10
1/0.1	no	no	10
		yes	10
	yes	no	10
		yes	10
10/1	no	no	10
		yes	10
	yes	no	10
		yes	10
None (no crosslinking)	no	n/a (not applicable)	10
	yes		10

## Data Availability

The raw computational datasets analyzed in this article are not readily available due to being part of an ongoing study, as well as technical and size limitations/restrictions. Requests to access the datasets should be directed to the authors Kerry Bloom and M. Gregory Forest.

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
