# Peer review of "The Role of Transient Crosslinks in the Chromatin Search Response to DNA Damage"

_ijms, 2025, doi:10.3390/ijms262311697_

Round 1

Reviewer 1 Report

Comments and Suggestions for Authors

Although this paper presents some potentially interesting findings regarding how crosslink depletion influences chromatin mobility and homology search, the clarity of writing makes it difficult to follow the logic, and the significance and applicability to real biological systems remain unclear. The authors should aim to simplify and clarify their descriptions, reduce jargon, and explicitly connect their modeling results to known mechanisms in yeast and (where appropriate) mammalian cells. Doing so would make the work more accessible and allow its contributions to be better appreciated.

Abstract:

  1. “Lossless” is not a standard scientific term in DNA repair. Perhaps the phrase that the authors were looking for was error-free repair or high-fidelity repair.? “Lossless” is confusing and not appropriate.
  2. The entire abstract is very confusing and not well written grammatically and clear. For example: “we explore the effects of inducing damage within a chromosome segment that, previous to the damage event, occupies a cross-linked region organized by transient structural maintenance of chromosome (SMC) complexes.” And later “synchronized with crosslinking deficiency of beads proximal to the break site.” Neither of these sections are clear as written.
  3. In addition, further down in the abstract, the authors state “We show that while the rate and density of dynamic binding tune the ability of the damaged site to explore space…”This statement is problematic as it is not the damaged site that “explore’s” the space. The authors should rephrase to avoid anthropomorphic or misleading wording.

Introduction

4. The sentence (lines 49-52) “Particularly, chromatin proximal to break sites demonstrates increased mobility with yeast cells exhibiting coordinated movement and displacement of the damaged segments to the outside of encompassing topological domain” is unclear and difficult to understand. The phrasing mixes multiple ideas (proximal chromatin mobility, coordinated movement, displacement to outside of domain) without clearly linking them. It also lacks precision—what exactly is meant by “coordinated movement” or “outside of encompassing topological domain”? This should be simplified and restructured into shorter, clearer statements.

5. Line 54: The phrase “cancerous mutations” is non-standard and unscientific. Mutations are not inherently “cancerous”; rather, they may occur in oncogenes or tumor suppressor genes and thereby contribute to carcinogenesis. A more accurate phrasing would be “oncogenic mutations,” “mutations that drive carcinogenesis,” or “mutations in cancer-associated genes.”

6. Figure 1: This figure is supposed to be a schematic overview of the simulation pipeline, but the legend is overly technical, long, and hard to follow. Instead of clarifying, it is overwhelming. Some of the issues include:

    1. The text dives into a multitude of things (tethering sites, bead colors, Rabl configuration, etc.), but it’s hard to tell what to focus on. For an “overview” figure, the main message (which I believe is crosslinks hold DNA in place, and depletion around a DSB frees the region) is lost.
    2. Although ROX” is introduced in the text, the legend should explain the acronym. Also “point-spread function” (PSF) should be explained as most won’t know what this means in a this context without explanation.
    3. It’s not clear why a PSF and “simulated microscope image” are relevant, and the legend should explain this.
    4. The explanation about vertex density, boundaries, and distances might be better in the Methods section as the legend should only explain what’s being shown (bead inside vs. outside of ROX boundary).
    5. The legend never explicitly tells the reader why this figure matters. It would be clearer and if it stated something like: “Together, these panels illustrate how the simulation defines chromatin chains, introduces a DSB with local crosslink depletion, and determines whether beads move inside or outside of the ROX domain.”
  1. Figure 4: The legend needs more information. It only says “Distances between the two beads immediately proximal to the break site” and leaves out critical details. Readers cannot interpret the figure without digging into the main text.
    1. The panels have gray highlighted cross-linking rates (slow, intermediate, fast), then with and without break, but this should be explicitly mentioned in the legend.
    2. The number of runs and statistics (average over 10 runs; shaded areas = standard deviation across runs) should be noted in the legend.
    3. The main finding, without a DSB (A–C), bead distances stay constant (15–45 nm, not sensitive to depletion). With a DSB (D–F), depletion allows ends to separate substantially, particularly at faster binding kinetics. Without these details, the figure cannot stand alone. 
  1. Figure 5: This figure and it’s legend are very difficult to follow. Some issues that stand out:
    1. The term “Contact maps depicting 3D bead-to-bead distances” seems strange. Contact maps usually depict contact frequency or probability, not distance. If this is truly distance, the term “contact map” may be misleading.
    2. The legend tries to explain four timepoints × four binding-rate conditions × multiple random seeds all at once, which is overwhelming. It would help to split into panels or sub-figures, with simpler captions.
    3. The color bar is labeled in nm (distances), but it’s not clear how to interpret darker vs. lighter patches (is darker = closer, lighter = farther?). The legend should explicitly say.
    4. The X/Y axes are said to correspond to bead numbers (1–361), but this isn’t obvious on the plots, and the mapping to chromosome XII (beads 61–421 of arm 2) is buried in the text. This should be clearer.
    5. The “yellow cross” that supposedly indicates separation of damaged DNA is very difficult to discern in most panels and the dashed boxes/crosses are not explained well in the legend.
    6. The legend says local depletion of crosslinks begins at 2500 s, but some panels are before and some after this point. It would help to mark on the figure which are pre- and post-depletion.
    7. The figure should state upfront: “These maps show that when crosslinks are depleted, the damaged region separates from the rest of the ROX, visible as a bright cross pattern that increases over time.” Then describe the details after this.
  1. Lines 519-527, The section beginning “Upon induction of a DNA break via activation of a dicentric chromosome…” through “…most likely reflects the fraction of DNA that is damaged” needs clarification. As written, it is difficult to follow and has logical inconsistencies. Some of the issues include:
    1. Terminology such as “expelled” and “exits” are non-standard. More precise phrasing (like “released from cross-linked regions” or “relocated outside the pericentromeric/nucleolar compartment”) would be clearer.
    2. The logic of comparing 22% (pericentromere) to 50% (nucleolus) is not clearly explained. It is not evident why this difference “most likely reflects the fraction of DNA that is damaged.”
    3. If the claim is that damaged DNA relocates out of constrained nuclear regions to increase accessibility to repair machinery, then the authors need to explain the biological trigger (why the loss of SMC crosslinks, due to chromatin remodeling?).
    4. The text should explicitly state whether this expulsion has been experimentally observed at all, or if it is purely a modeling inference. Right now, it’s not clear how the simulation results are linked to real yeast or mammalian cell biology.

Discussion

  1. The first sentence (line 465) “The organization of chromosomes in the nucleus leads to a profound problem in information access” is confusing and misleading. Chromosome organization is not itself a “problem.” The likely intended meaning, suggested by the following sentence, is that the nucleus is highly compacted, making it challenging for repair machinery to access DNA sequences. The discussion should open with a clearer framing, e.g., “Because the genome is densely packed within the nucleus, access to DNA for processes such as repair and transcription is constrained.”
  2. The sentence (Lines 526-7) “The lower fraction of DNA that exits the pericentromere (22%) relative to the 50% of the damaged DNA that exits the nucleolus most likely reflects the fraction of DNA that is damaged” is confusing as written, and not clear:
    1. What is meant by “DNA exiting the pericentromere” or “exiting the nucleolus”?
    2. How these percentages (22% vs. 50%) were measured or defined.
    3. Why the difference would “most likely reflect the fraction of DNA that is damaged.”

This section would be improved with further clarification, and by explicitly stating what is being compared (damaged vs. total DNA? different nuclear compartments?), how those fractions were determined, and why that difference reflects DNA damage.

  1. The final sentence of the discussion (lines 546-549) is unclear and somewhat contradictory: “While there may be active mechanisms that drive DNA motion, the release of cross-links is sufficient to expel DNA from the chromosome territory and increase its range of motion. This study highlights another powerful organizational feature that emerges from the understand the relevance for what is happening in yeast and mammalian cells…”

a) The phrasing is confusing and unfinished (“emerges from the understand”).

b) It is also important to explain why this would be biologically relevant, whether SMC depletion is a modeled event, or whether there is experimental evidence that crosslinks are actively removed during damage in yeast (and by analogy, mammalian cells).

Without this the conclusion sounds somewhat circular: removing SMC crosslinks in the model obviously leads to DNA being less constrained, so what is the broader biological insight?

Comments on the Quality of English Language

Please see in main body of comments to the authors in my review.

Reviewer 2 Report

Comments and Suggestions for Authors

This manuscript addresses an important and timely question in genome biology: how chromatin organization, specifically transient SMC-mediated crosslinks, regulates the mobility of double-strand break (DSB) sites and thereby influences the homology search during repair.

Specific comments:

  1. While the yeast dicentric chromosome assay is informative, the study relies on one main experimental setup. Additional evidence would strengthen the generality of conclusions.
  2. The manuscript attributes chromatin displacement largely to SMC crosslink depletion, but the biological mechanism of depletion after DSBs is not fully addressed. Are histone modifications or checkpoint kinases involved?
  3. The imaging results are largely described qualitatively. More rigorous statistical analysis would be helpful.
  4. The polymer model makes strong assumptions. Discussion of limitations and how parameter choices might bias results should be expanded.

Reviewer 3 Report

Comments and Suggestions for Authors

Review of Manuscript entropy-3868399

Journal: Entropy
Title: The role of transient crosslinks in the chromatin search response to DNA damage

            In this study, the authors have modeled the movement of DNA after a double-strand break is introduced at a locus that is analogous to a chromosome region where SMC protein complexes may bind and generate cross-links. Several variables related to timing and distance and density were analyzed. In addition to the in silico experiments, special dicentric strains of budding yeast were constructed and the effects of breakage near a natural centromere were analyzed. The authors are to be commended for combining in silico data with yeast cell experiments and comparing them. The paper is largely well written, but clarification is needed for portions of the manuscript.

Items:

  1. Lines 349 - 356: "Plots A-F show 3D Euclidean distances between beads 240 and 241 over time, with 349 each row corresponding to a different crosslinking regime and each column correspond-350 ing to a different DSB setting." I was not able to understand what rows and columns of what figure were being referred to in this paragraph and some clarification is needed.

  1. Lines 405-407: "These results suggest that crosslinking spatiotemporal dynamics in the ROX can regulate the motion of the damaged ends proximal to a DSB, thus influencing homology search and the repair pathway." The paper refers to HR here and elsewhere, and this makes sense since it is the primary pathway in the yeast cells used for the experimental section. Should there be some mention of relevance to repair by NHEJ too, as it is the dominant DSB repair pathway in higher eukaryotes and is also present in yeast?

  1. Lines 445-454: This paragraph begins analysis of Figure 6 in the Results and refers to one of the three major parts of the figure but not the table and the images. The figure's results are partially described in the Discussion section in lines 519-531. It would be more normal to report the data in Figure 6, especially the table, in the Results section and it would naturally flow from line 454 at the end of the Results. Just as important, I didn't see the rad52 mutant results in the Figure 6 table described directly in the text (results of a past rad52 study were mentioned in the Discussion but not the current rad52 data) and these should be reported.

  1. Line 442 (legend of Figure 5): "...i.e., beads (61–421) of arm 2 of chromosome XII." Should a specific reference to yeast chromosome XII be here given that the experiment was a bead simulation...

  1. When performing computational studies it is normal to have to make both reasoned and arbitrary decisions about numerical cutoffs and other numbers used in the analysis. Accordingly, many choices had to be made here, e.g., defining ROX boundaries based on 91-94%, "361 beads in the ROX", 20 beads in a damaged region, 10 simulations or 40 total simulations, N=10 runs, 1300 total seconds, etc. I cannot judge the appropriateness of all the decisions that were made and would simply ask the authors to please confirm again during revision the validity/justifications used for each number to be confident of their usefulness and robustness.

  1. Line 24 (Abstract) - "We show that while the rate and density of dynamic binding 23 tune the ability of the damaged site to explore space, the type and extent of damage is the 24 key driver of local dynamics." My understanding is that the focus of the paper is on a DSB. Is it correct to refer to the "type" of damage as a variable here if the study was focused on DSBs and other types of damage were not examined.

  1. Line 34, "The majority of DSBs occur programmatically and are managed by a set of 34 repair responses whose regulation plays an essential role in cell survival." Should "majority of DSBs" be changed to simply "Many DSBs" here, since most DSBs experienced in the life of a cell are not programmed DSBs.

  1. Line 533: "...global changes to the material property". The meaning of material property is unclear and it may be best to adjust the sentence for clarity.

Round 2

Reviewer 1 Report

Comments and Suggestions for Authors

The authors have addressed the issues.

Reviewer 2 Report

Comments and Suggestions for Authors

No further comments at this stage.